# Protein Hydrogels: The Swiss Army Knife for Enhanced Mechanical and Bioactive Properties of Biomaterials

**DOI:** 10.3390/nano11071656

**Published:** 2021-06-24

**Authors:** Carla Huerta-López, Jorge Alegre-Cebollada

**Affiliations:** Centro Nacional de Investigaciones Cardiovasculares (CNIC), 28029 Madrid, Spain

**Keywords:** hydrogel, protein, mechanical modulation, viscoelasticity, extracellular matrix, folding, single-molecule, nanomechanics

## Abstract

Biomaterials are dynamic tools with many applications: from the primitive use of bone and wood in the replacement of lost limbs and body parts, to the refined involvement of smart and responsive biomaterials in modern medicine and biomedical sciences. Hydrogels constitute a subtype of biomaterials built from water-swollen polymer networks. Their large water content and soft mechanical properties are highly similar to most biological tissues, making them ideal for tissue engineering and biomedical applications. The mechanical properties of hydrogels and their modulation have attracted a lot of attention from the field of mechanobiology. Protein-based hydrogels are becoming increasingly attractive due to their endless design options and array of functionalities, as well as their responsiveness to stimuli. Furthermore, just like the extracellular matrix, they are inherently viscoelastic in part due to mechanical unfolding/refolding transitions of folded protein domains. This review summarizes different natural and engineered protein hydrogels focusing on different strategies followed to modulate their mechanical properties. Applications of mechanically tunable protein-based hydrogels in drug delivery, tissue engineering and mechanobiology are discussed.

In this review, we present an overview on different protein hydrogels with optimized biological and mechanical properties, and their current applications in basic science and biomedicine. We will review evidence supporting the mechanical design of protein hydrogels based on the nanomechanics of the hydrogel building blocks. In combination with their well-known bioactive properties, protein hydrogels emerge as all-purpose, multifunctional biomaterials, with a versatility that resembles that of a top-of-the-line Swiss army knife. To provide a better perspective on the topic and highlight the advantages of protein hydrogels, we introduce first the wider field of hydrogels. This field is vast, and we apologize for missing relevant references that may have escaped our scrutiny.

## 1. Hydrogels

Hydrogels are three-dimensional hydrophilic polymeric networks swollen in large quantities of water that can respond to environmental stimuli, such as pH, temperature, ionic strength and electric fields [1]. When swollen, hydrogels are soft and rubbery and resemble living tissues. Their high-water content and desirable properties make them ideal candidates to explore biological and biomedical applications [2]. Hydrogels were first described in 1960 by Wichterle and Lim [1], moment in which their use was very limited by concerns of toxicity of required crosslinking agents and inability to operate within physiological conditions. Since their discovery, research has allowed the design of more biocompatible hydrogels that can be employed as biosensors, drug delivery carriers and implant scaffolds [3,4,5].

Hydrogels can be classified according to different parameters: physical structure, network electric charge, type of crosslinking, composition and chemical nature, as reviewed elsewhere [6]. Based on physical structure, hydrogels are either semi-crystalline or amorphous, and their electric charge classifies them as ionic or neutral [7]. There are two main types of crosslinking strategies employed in hydrogel production: chemical and physical [8]. Chemically crosslinked hydrogels result from covalent bonds between the chains in the hydrogel. These chemical bonds control the degree of swelling of the hydrogel. Physically crosslinked hydrogels result from non-covalent interactions such as ionic bonds, hydrogen bonds, or molecular entanglements. This type of crosslinking can be reverted by application of mechanical force or other environmental changes. Polymeric composition classifies hydrogels as homopolymeric, copolymeric or multipolymeric [3,7]. Homopolymeric hydrogels, originate from just one specific class of monomer, whereas copolymeric hydrogels are composed by two or more distinct classes. Multipolymeric hydrogels include both Interpenetrating (IPNs) and Semi-Interpenetrating Networks (Semi-IPNS). Multipolymeric hydrogel assembly involves the polymerization of two or more polymers; usually one of them is already pre-polymerized and placed into a solution of monomers of the second polymer and a polymerization initiator. The reaction can take place in the presence of a crosslinking agent, in order to form a complete IPN, or in the absence of the crosslinking initiator to form a Semi-IPN [7,9,10]. Lastly, based on the nature of constituent polymers (i.e., chemical nature), hydrogels can be classified as either natural, synthetic or hybrid. Among the different subsets of polymeric hydrogels, this review focuses on protein-based hydrogels, a highly versatile type of biomaterials that have flourished in the last two decades showing great promise in several biological and biomedical applications.

## 2. Mechanical Properties of Hydrogels

Applications of hydrogels must take into account their mechanical properties. These include stiffness, energy dissipation or viscosity, plasticity, yield and ultimate strength among others [11]. The most commonly performed test to measure the mechanical properties of materials involves unidirectional mechanical load. The application of load generates stress (σ), which is defined as force per initial unit area, on the material. In unidirectional tests, material deformation in response to force is called strain (ε), which is defined as the change in length of the specimen from its gauge length to the final length [12]. 

There is a number of standard tests used to characterize the mechanical properties of materials, such as stress-strain, stress-relaxation and creep tests [13,14,15]. These tests give useful information on whether the behavior of the material matches that of an elastic, viscoelastic or viscoplastic solid. Classical elastic solids show a linear stress–strain relationship given by σ = Eε, where the slope E is a constant called Young’s modulus or elastic modulus, which indicates the stiffness of the material [12] (Figure 1b). When elastic materials are subject to cyclic deformations, no energy is dissipated, as can be seen by the overlapping forward and backward stress–strain curves (Figure 1a). Furthermore, both stress and strain responses are instantaneous and time-independent (Figure 1b) [13]. 

Viscoelastic behavior involves both elastic solid and viscous fluid response [13]. The behavior of linear viscous fluids is described by a relationship between stress and strain rate of the form σ = μė, in which μ represents the viscosity (Figure 1c). The material never returns to its original shape, the strain is permanent. During this series of histories, the work is completely converted to thermal energy. In viscous fluids stress and strain do not show a proportional response, but straining continuously takes place with time when a constant stress is applied, highlighting the time-dependency of fluid responses (Figure 1d). In viscoelastic polymers, the presence of crosslinks causes the network to recover its original shape, making the process reversible but time-dependent, not immediate. Time-dependency can be identified as hysteresis resulting from energy dissipation in stress–strain curves (Figure 1e). Macroscopically, the application of a constant stress to a viscoelastic solid material results in an instantaneous increase of the strain (elastic) followed by continuous straining in time at a non-constant rate (viscous). Thus, removal of stress leads to some instantaneous recovery of strain (elastic) that is followed by a delayed recovery (viscous) (Figure 1f). Elastic deformations -let them be viscous or not- are reversible, meaning the initial mechanical properties are completely recovered upon load removal. In contrast, plastic strains involve the irreversible deformation of the material. Almost all real materials will undergo plastic deformation to some extent [16,17]. Materials with an initial viscoelastic response that undergo plastic deformations are referred to as viscoplastic. In viscoplastic materials there is always some residual strain that cannot be recovered (Figure 1g,h).

## 3. Composition and Structural Properties of Hydrogels

The composition and structure of hydrogels are also determining factors in the set of applications hydrogels can participate in [18]. Three parameters can define the structure of hydrogels: the polymer volume fraction in the swollen state, the average molecular weight between crosslinks, and the pore size [19]. Hydrogel structure and mechanical properties are interconnected. The persistence length of the building blocks and their cross-sections also represent physical parameters that provide information of the mechanical properties of a hydrogel. For example, the larger the persistence length the stiffer the fibril [20].

There are several case studies concerning hydrogels where physical features have been linked to mechanical response. For instance, introduction of amino acid changes in coiled coils in engineered protein hydrogels tunes the degree of self-assembly from dimers up to heptamers [21]. These changes in coiled-coil structure effectively affect crosslinking density, and thus, mechanical response. At the same time, the strength and stiffness of silk are correlated with the number of β-sheet crystallites and with their orientation within fibrils [22]. Strategies involving changes in amino acid sequence to favor the interaction between negatively and positively charged amino acids to enhance co-assembly, as well as lengthening the number of total amino acids in protein building blocks, have been followed to tune the mechanical response of the hydrogels by promoting stiffness [23,24]. 

The structural features of polymers can be determined through a variety of techniques, such as Small Angle Scattering (SAS) methods. Small Angle X-ray Scattering (SAXS) is a universal technique used to study the structure of noncrystalline systems on a nanometer scale [25,26]. For biological objects, analysis of the low-resolution structure of macromolecules and their complexes, including hydrogels, is among the most important SAXS applications [27]. SAXS is particularly useful because it allows the investigation of samples in their wet-state without the need for any sample preparation such as drying and/or freezing, although scattering patterns require theoretical models in order to be interpreted [28]. SAXS can be employed to analyze the kinetics and gelation mechanism of self-healing hydrogels with different dynamic interactions. Moreover, SAXS can be used to assay the effects of pH, temperature and shear on the structure-mechanics relationship of hydrogels. Examples include chitosan-based hydrogels [29] and lamellar hydrogels from microbial glucolipids [30], where thanks to SAXS it was possible to link structural changes and the rheological (mechanical) properties of the hydrogels. 

Measurements of hydrogel ultrastructure can also be obtained using Small Angle Neutron Scattering (SANS). This technique differs from SAXS on the degree of penetration through the sample [31,32]. Neutrons generally penetrate several centimetres through most materials, whereas X-rays penetrate only tens of micrometres. The high penetration of neutron beams gives them unique advantages in many applications, where it can be used to detect very small structures, like crystallites in poly(vinyl alcohol) hydrogels [33] or amorphous associations and their dependence on polymer length in poly(ethylene glycol) hydrogels [34]. Interestingly, SANS can also be applied to the detection of nanostructural changes associated with energy dissipation [35], which has the ability to give essential information on the molecular events taking place during stress-relaxation in hydrogels. 

## 4. Protein Hydrogels

Protein-based hydrogel biomaterials have emerged as an attractive alternative to classical polymeric hydrogels due to their particular properties [8]. Protein hydrogels are a type of polymeric materials that use proteins as their building blocks. Proteins possess diverse genetically encoded structures and function. Consequently, one of the most remarkable characteristics of protein hydrogels is the preservation of the properties inherent to their protein components [36]. Moreover, most protein hydrogels are inherently biologically friendly and biodegradable [8]. 

### 4.1. Types of Protein Hydrogels

Originally, protein hydrogels were strictly made of proteins that naturally crosslink or gelate, such as elastin and collagen [37,38], but the evolution of molecular biology and protein biochemistry methods has made it possible to develop a broad variety of engineered protein-based materials. Therefore, protein hydrogels can nowadays be generated using natural proteins isolated from animal or vegetal sources, synthetic proteins or a combination of both [6,8,37,39]. 

#### 4.1.1. Natural-Protein-Based Hydrogels

The main natural proteins used to generate hydrogels include collagen, gelatin, elastin, laminin, fibrin, silk fibroin and globular proteins such as lysozyme, BSA and ovoalbumin [40,41,42]. Collagen is the main extracellular matrix (ECM) protein, where it provides mechanical support to tissues. Its complex quaternary structure accounts for up to 29 different collagen types, being collagen I the most commonly employed in hydrogel production [43,44,45]. Collagen is biodegradable, presents low antigenicity and low inflammatory response [46,47]. Some of the limitations of collagen hydrogels are the thrombogenic potential of collagen’s degradation products, the high cost of pure collagen and the lack of sufficient mechanical strength, which needs to be compensated via additional physical or chemical crosslinking [48]. 

Gelatin is a polymer obtained through the denaturation of collagen; a low-cost, barely immunogenic mixture capable of undergoing a reversible sol-gel transition below room temperature [49,50,51]. Gelatin hydrogels show poor mechanical properties and often require extensive crosslinking for many of their applications [46]. 

Elastin is an insoluble ECM protein that provides various tissues in the body with the properties of extensibility and elastic recoil [52]. Elastin-based biomaterials are increasingly applied due to their remarkable properties such as elasticity, long-term stability, and biological activity [53]. Elastin is insoluble as consequence of extensive lysine crosslinking, and therefore difficult to process. Consequently, soluble forms of elastin including α-elastin, an oxalic acid-solubilised derivation of elastin [54,55], and tropoelastin, the soluble precursor of elastin [56,57], are frequently used to form crosslinked hydrogels. 

Laminin is a heterotrimeric glycoprotein with a key role in the modulation of neural stem cell (NSC) behavior, including cell adhesion and viability [58,59]. Hence, laminin is highly attractive for the design of NSC niche microenvironments. 

Fibrin is a blood protein involved in tissue repair and coagulation. Fibrinogen is its inactive form and when activated it forms fibrin networks [46]. When used as a substrate, fibrin materials allow cell growth and better ECM deposition than other natural protein-derived hydrogels [46,60]. Its main inconvenient is the lack of mechanical strength and its high degradability, which is why it is often combined with protease inhibitors or other components such as polydimethylsiloxane (PDMS) in hybrid hydrogels [61]. 

Silk fibroin is a protein produced by silkworms, spiders and scorpions. It possesses excellent mechanical properties, low adverse immune reaction, minimal thrombogenicity, and compatible degradation rates [62,63]. Furthermore, it is the most versatile amongst the natural proteins used in hydrogel formation, being compatible with several manufacturing processes like 3D printing technology and lithography [46,64]. 

Another hydrogel widely used in cell biology studies is Matrigel (BD Biosciences, Mississauga, Canada), a gelatinous, complex protein mixture derived from mouse Engelbreth-Holm-Swarm tumors, which contains mainly laminin, collagen IV and enactin [65]. In vivo, it is used to improve graft survival, repair damaged tissues, and increase tumor growth [66]. Despite its extensive use as cellular matrix, just like the rest of natural protein-derived hydrogels, Matrigel lacks control of mechanical properties and suffers from lot-to-lot variability.

#### 4.1.2. Engineered Protein-Based Hydrogels

Pioneered by the Tirrell and the Kopecek groups, the field of synthetic protein hydrogels has grown rapidly over the past two decades thanks to significant progress in recombinant DNA technology and protein engineering techniques [8,67,68]. These materials use designed recombinant proteins as building blocks [69]. The resulting biomaterials generally possess enhanced mechanical properties and increased batch-to-batch reproducibility than hydrogels made from natural proteins [8,70]. Engineered protein hydrogels can be built entirely with synthetic proteins or with a hybrid network of proteins and other components. As recombinant proteins are polymers with well-defined sequences and folded structure, it is possible to fine-tune the properties of protein hydrogels by genetically mutating the protein building block [71]. From a mechanics point of view, recombinant proteins can be designed to adopt unique secondary and tertiary structures, which are responsible for their mechanical properties [70,72]. 

Elastomeric proteins are a type of mechanical proteins found in nature that have inspired numerous biomimetic protein polymers used in hydrogels [73]. One of the first elastomeric proteins engineered into hydrogels was resilin, which adopts a random coil structure and functions as an entropic spring [74]. Elastin is another example of a natural elastomeric protein from which synthetic hydrogels have been engineered [75]. Elastin-like proteins (ELPs) are produced by recombinant protein synthesis. These soluble proteins are designed by mimicking the useful functionalities found in elastin. In ELPs, repetitive short elastin-like structural sequences confer elasticity and resilience to strong and tough hydrogels that hold the ability to self-heal [76,77]. Similar to classical polymer hydrogels, protein hydrogels can be produced following chemical and physical crosslinking strategies. A typical chemical crosslinking involves the photochemical formation of dityrosine crosslinks, which is inspired by the reaction first reported by Fancy and Kodadek [78]. Several works led by the production of resilin-based biomaterials [74,79] have followed this strategy to crosslink proteins into hydrogels. The group of Hongbin Li adhered to this type of crosslinking and found the way to produce muscle-mimicking hydrogels based on modular constructs containing the elastomeric protein GB1 (the streptococcal B1 immunoglobulin-binding domain of protein G [80]) and resilin [81] and to modulate their mechanical properties by varying the crosslinking density [82].

In addition, novel, more efficient or practical methods are constantly being developed [8]. For instance, SpyTag-SpyCatcher chemistry can be used to chemically crosslink tandem modular elastomeric proteins at room temperature resulting in soft hydrogels [83,84]. SpyCatcher/Tag crosslinking is based on the spontaneous formation of a covalent isopeptide bond between two split peptide fragments derived from the CnaB2 domains of Streptococcus pyogenes [84,85,86]. Moreover, SpyTag-SpyCatcher chemistry can also be used to selectively decorate a “blank slate” pre-existing hydrogel. For example, modular constructs containing the GB1, resilin and SpyCatcher can be chemically crosslinked into hydrogels, which can later be decorated with molecules conjugated to SpyTag [87]. The SpyTag-SpyCatcher decorating strategy allows to modulate the biochemical behavior of the crosslinked hydrogels without altering hydrogel mechanics [88]. Proteins can also be combined with synthetic molecules like PEG via Michael addition to enhance and tune the mechanical response of the resulting hydrogels with temperature and ionic strength changes [89]. 

## 5. Bottom-Up Mechanical Design of Protein Hydrogels: Lessons from Titin

As discussed above, one of the advantages of protein hydrogels is the possibility to fine-tune their macroscopic properties in a rational fashion. Hydrogel mechanics are included amongst these relevant properties as the mechanical behavior of engineered hydrogels is essential to determine their target applications [90,91,92]. An emerging approach is to mechanically engineer protein-based hydrogels starting from the protein building blocks, in what it is known as a bottom-up approach, from proteins to materials (Figure 2). 

Pioneering work by the group of Hongbin Li got inspiration from the mechanical function of the muscle protein titin to show that the nanomechanical properties of proteins could be scaled up to the macroscopic mechanical properties of hydrogels [81] (Figure 3). This milestone contribution showed that it was possible to rationally engineer novel hydrogels with optimized mechanical properties building on the nanomechanics of the protein building blocks, which were studied through single-molecule force spectroscopy techniques based on atomic force microscopy (AFM) [73,93] (Figure 2; Figure 3). Nowadays, researchers are looking for engineered proteins with mechanical properties that mimic or even surpass those of natural ones, in order to use them to produce protein hydrogels [73]. Additionally, taking inspiration from protein hydrogels, polymer-based hydrogels have been designed to directly correlate single-molecule and bulk mechanical properties [93].

Elastomeric proteins can be classified as entropic-spring-like and shock-absorber-like [94]. Entropic elastomeric proteins like resilin or elastin are made of flexible, non-globular, and often unstructured polypeptide regions, while shock-absorber-like proteins consist of individually folded globular domains, which are typically arranged in tandem. Titin, the largest protein encoded by the human genome (363 exons and over 30,000 amino acids in humans), is the natural elastomeric protein whose mechanical properties have been most extensively studied [94,95,96,97,98,99]. Titin is one of the main filaments of the sarcomere, the basic contractile unit of striated muscle [100] (Figure 4). In the sarcomere, titin is a major contributor to stiffness and energy dissipation [96]. Titin spans half the length of the sarcomere, from the Z-disk to the M-line (Figure 4a), and it adjusts its total length at the I-band of sarcomeres to the needs of the working muscle [101,102]. In its sequence titin includes both entropic-spring-like segments and shock-absorber, folded domains.

The mRNA coding for titin is alternatively spliced, resulting in muscle-specific isoforms [103]. These isoforms differ in their length and the ratio of folded domains to unstructured regions in the I-band, resulting in titin molecules with tailored mechanical properties [96]. For instance, the I-band of titin N2B adult cardiac isoform can be subdivided into four structural regions: a proximal immunoglobulin-like (Ig) region containing 15 tandem Ig domains; an entropic middle 572-amino-acid-long unstructured N2Bus segment; an entropic 186-amino-acid-long PEVK segment (rich in proline, glutamate, valine and lysine residues); and a distal Ig region containing 22 tandem Ig domains [97]. 

**Figure 4 nanomaterials-11-01656-f004:**
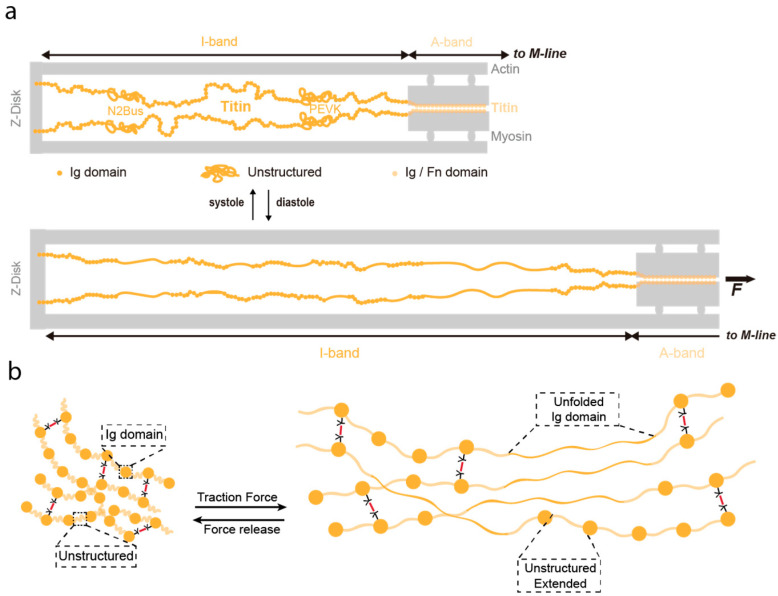
Biomaterials inspired in the mechanical function of titin. (**a**) Schematic depiction of one half sarcomere (not to scale). Titin is colored in yellow, while other sarcomeric proteins appear in grey. Immunoglobulin-like domains are represented as filled circles, and the approximate positions of the unstructured N2Bus and PEVK domains are indicated. The length of the mechanically active I-band and the beginning of the A-band are delimited by arrows. The top part represents a contracted sarcomere (e.g., heart systole). The bottom part represents an extended sarcomere (e.g., heart diastole) where unstructured regions have extended and a fraction of Ig domains have unfolded. Reproduced with permission from [104]. Copyright, Elsevier, 2019. (**b**) Titin-based hydrogel built with Ig-like domains and unstructured regions that can undergo the same reversible extension and unfolding as titin in the sarcomere.

As can be deduced from the domain organization in titin, two complementary mechanisms contribute to its mechanical response: purely elastic extension of PEVK, N2Bus and the tandem Ig domains, and dynamic unfolding/refolding of the Ig domains (Figure 4a). When an Ig domain unfolds under force, around 100 amino acids are released, increasing by approximately 30 nm its contour length. Hence, the protein domain becomes softer, whereas refolding has the opposite effect. The balance of folded versus unfolded domains is a key contributor to the overall mechanical behavior of titin [104,105]. This balance can be modulated through different mechanisms, including the induction of posttranslational modifications (PTMs) on cryptic cysteine residues [104,106]. For instance, S-thiolation of titin domains block protein folding, resulting on titin softening [107]. In contrast, formation of intradomain disulfide bonds in titin domains results in stiffening and can lead to mechanical adaptation through isomerization reactions [108,109]. In addition, point mutations can mechanically stabilize or destabilize Ig domains, hence, making them stiffer or softer respectively [110]. Lastly, another well-known regulator of protein mechanics is pulling geometry, because the mechanical resistance of a globular protein is highly dependent on the direction force is applied [111,112,113,114,115]. 

The array of single-molecule biophysics studies performed to understand titin’s mechanical properties and general mechanical behavior [78,95,105,110,116,117,118] have highlighted its essential role in muscle homeostasis. Its outstanding properties have turned titin into a reference when it comes to designing building blocks to engineer hydrogels with enhanced mechanical properties. In the pioneering design by the group of Honbing Li introduced above, hydrogels were cast by chemically crosslinking the elastomeric proteins GB1 and resilin and, just like titin, they behaved as entropic spring-like materials at low strains and as shock-absorbers at high deformations [81]. In a simple interpretation that mirrors what is known about titin mechanics, resilin residues behave as purely entropic strings, while mechanical unfolding and refolding transitions of GB1 domains contribute to stiffness by setting the fraction of unfolded polypeptide regions. Unfolding and refolding of GB1 domains would also lead to energy dissipation in the resulting hydrogels (Figure 4b). The mechanical response of the hydrogels could be modulated by changing the proportion of resilin regions, as it would happen during titin isoform splicing, although the obtained modulation did not always correlate with the proportion of resilin [81]. These discrepancies may originate from the concomitant alteration of crosslinking sites, which were present in both GB1 domains and resilin regions. Hence, although the development of these titin-mimicking biomaterials opened the door to the targeted bottom-up mechanical design of protein hydrogels (Figure 3), which so far have been produced also from alternative building blocks such as the I91 domain of titin [119,120] or protein L [121], the rational design of protein hydrogels with independently controllable mechanical properties remains incomplete. To truly accomplish this goal, it is critical to understand the link between the mechanical properties of individual proteins and the resulting hydrogels. In this regard, recent work has demonstrated that carefully designed protein building blocks can be combined to synthesize protein hydrogels with predictable, although extreme, mechanical properties [122], and that hybrid polymer-protein hydrogels can take advantage of protein extensibility under force to achieve remarkable anti-fatigue fracture [123]. Theoretical models that account for the emergent mechanical properties of protein building blocks, all the way from single-molecules to the hydrogel mesh, have been proposed [122,124,125]. Accurate models of this sort will be necessary to fully achieve the rational mechanical engineering of protein hydrogels.

## 6. Smart Protein Hydrogels

Unlike conventional hydrogels, smart or stimulus-responsive hydrogels enable post-synthesis modification of their properties, which can be triggered by environmental factors that promote changes in internal organization [126,127]. There are many types of triggers, like pH [128], temperature [129,130], ionic strength [131], electric fields [132] and light [133]. Smart protein hydrogels constitute a subgroup of engineered protein hydrogels, as protein building blocks can be designed to exhibit structural transitions at the molecular level in response to external stimuli. Response to pH is frequently implemented in protein hydrogels. For example, a PEG-polypeptide hydrogel can be used to encapsulate larger proteins so that cargo release is pH-dependent [128]. 

Another class of responsive protein hydrogels involves self-assembly in response to a complementary protein or particle to create composites. This physical recognition enables modulation of mechanical and structural properties during the assembly process as well as specific self-healing after stress application [134,135,136,137]. For example, modular proteins containing an elastomeric domain and a leucine zipper domain are able to self-associate into hydrogels and thermo-reversibly transit back to solution form at temperatures over 60 °C [138]. Improvement of the leucine zipper system was achieved by fabricating two complementary leucine zipper sequences that could be produced independently, in order to avoid spontaneous self-association during purification, and mixed at different ratios to change the degree of crosslinking [139]. 

It is also possible to produce protein hydrogels whose elastic modulus can be modified through the alteration of ionic strength [140,141], redox environment [119,120,136,142,143] or metal chelation [144]. In some of these examples, mechanical modulation is achieved through control of protein contour length and mechanical folding/unfolding dynamics. Recently, it has been demonstrated that shape changes in protein hydrogels can be achieved by adsorption of Cu^2+^ and Zn^2+^. This change in shape was linked to marked variations in stiffness (up to 17-fold), exceeding the current range of stiffness existing for protein hydrogels [142]. An alternative strategy is based on the modulation of protein folding in protein hydrogels containing polyelectrolytes [141].

Light-control of hydrogel mechanics is a long-sought goal due to the possibility of remote activation and high spatiotemporal control [145,146,147]. The use of light to modulate the properties of protein hydrogels has just begun and remains largely under-explored. Other responsive mechanisms include enzymatic modifications and/or degradation of hydrogels [148,149]. Although temperature can reversibly regulate hydrogel behavior, it is generally irreversible and may lead to protein aggregation in hydrogels [150]. Most protein-based thermal-responsive hydrogels are based on the reversible phase transition of ELP-based polypeptides [77,151,152]. Hydrogels can concomitantly respond to more than one stimulus. For instance, protein fragment reconstitution can be used to reversibly create and decorate a protein hydrogel responsive to both temperature and redox status [153]. 

This is just a short list of the wide spectrum of existing smart protein hydrogels, indicating the large versatility of this family of biomaterials. Of note, some of titin’s Ig domains -mainly I91, the best characterized one- have been included as building blocks of smart protein hydrogels [154] whose mechanical properties are tunable through temperature, redox modifications [119,120], alteration of the folded/unfolded ratio of domains [124], or metal chelation [144,155]. In principle, any mechanism that modulates the nanomechanics or protein building blocks, such as redox modifications of cryptic cysteines in folded domains, could be adapted to modulate hydrogel mechanics [104]. In the context of hydrogels, it is important to consider also potential contributors to their macroscopic mechanical properties that do not stem from protein nanomechanics, such as the stiffening effect induced by massive unfolding and aggregation of protein building blocks [150]. 

## 7. Applications of Protein Hydrogels

Due to their diversity and adaptability, protein biomaterials can accommodate a broad range of functional requirements, starting from their original range of application involving cosmeceutics and wound healing [156]. In fact, protein hydrogel formulations from natural, semi, or synthetic polymeric materials have gained great attention in recent years for treating various skin conditions and for cosmetology procedures. Among the natural proteoglycans and proteins used, collagen, fibrin, gelatin, keratin, silk fibroin, and eggshell membrane are particularly important [157,158,159]. There are several cosmeceutic formulations that contain collagen hydrogels and that are involved in the treatment of scars and wrinkles, such as CosmoDerm^®^, CosmpoPlas^®^, Fibrel^®^, Zyplast(R)^®^, Zyderm(R)^®^ and Evolence^®^ Collagen Filler [160,161,162].

Hydrogel dressings based on natural proteins are excellent tools for treating wounds because of their structural and mechanical features that make them resemble soft physiological tissue [156]. Protein-based hydrogels provide effective treatment for wounds of various origins. One of the most commonly used polymers in the production of hydrogels for wound healing is collagen, due to its biocompatibility, biodegradability, biological profile, and promising results both in vitro and in vivo [157]. Collagen also seems to be a good choice to form hydrogels for wound healing due to its ability to recruit specific types of cells to the wound site, absorb exudates, maintain a moist wound environment, and stimulate the healing process by deactivating excessive matrix metalloprotease [163,164]. A correct wound healing process can also be achieved using fibrin or keratin hydrogels [159,165]. Silk fibroin is another protein that can be used to create hydrogels with potential use in wound healing, even in third-degree burn wounds [166].

The physiologically relevant mechanical properties of protein hydrogels, as well as their degradability, enhanced functionalization and similarity with cell and tissue environments enable processes such as drug delivery and tissue engineering [167]. These same properties make protein hydrogels excellent candidates to explore how cells sense and react to ECM mechanics, of particular interest in the field of mechanobiology [158,168].

### 7.1. Drug and Cell Delivery

Conventional drug administration often needs repeated high-concentration doses to achieve a therapeutic effect, which is not efficient and can cause side effects [169]. Moreover, peptide and protein drugs usually have short half-lives of only minutes to hours [170,171]. Controlled drug delivery systems appear as an alternative addressing these limitations. Hydrogels are particularly appealing due to their biocompatibility and general versatility [172]. Silk is one of the most versatile proteins employed in biomedical applications. The group of David Kaplan came up with silk fibroin hydrogels to steadily and locally deliver murine monoclonal antibodies through hydrophobic/hydrophilic silk-antibody interactions [173]. This design was further employed to repair the maxillary sinus floor through the delivery of vascular endothelial growth factor (VEGF) and bone morphogenic protein-2 (BMP-2) [174]. Focusing on gene delivery, the same group bioengineered recombinant silk proteins containing poly(L-lysine) complexes to home specifically tumor cells using tumor-homing peptides [175]. Additional functionality may also be gained by producing silk-elastin-like polymers to achieve fine control of biodegradation rates and thus, control the delivery of plasmid DNA [176] and adenoviral vectors [177]. Collagen and gelatin are also commonly employed in drug delivery [178,179,180]. In fact, Infuse™ collagen hydrogel implants for BMP-2 and BMP-7 delivery have successfully made it to the clinic for the treatment of long bone fracture and spinal fusion [181]. Soy protein hydrogels can work as targeted delivery systems for molecules like riboflavin [182]. In these hydrogels, molecule release is triggered by changes in pH, which allow site-specific delivery. Hybrid and synthetic protein hydrogel systems are also employed in drug delivery. A B12-dependent photoresponsive protein hydrogel has been designed for controlled protein and stem cell release [133]. These hydrogels are entirely composed of recombinant ELPs, which are fused to an adenosylcobalamin binding domain (CarHC) using SpyTag-SpyCatcher chemistry. This system could tetramerize and cast a hydrogel in the dark and undergo a rapid gel-sol transition caused by light-induced CarHC disassembly. ELPs are indeed versatile elements in the design of protein hydrogels involved in targeted drug delivery. Their response to thermal changes makes them ideal carriers of radionuclides, chemotherapeutics and biomolecular therapeutics to tumors, which are specifically released in the tumor thanks to regional hyperthermia [151]. Another example is the use of the elastomeric protein GB1 combined with SpyTag-SpyCatcher technology to create hydrogels that work as cell and drug carriers [183]. It is also worth mentioning drug delivery applications based on tetratricopeptide repeats (TPR), helix-turn-helix protein structures that interact through a single inter-repeat interface to form elongated superhelices [184]. The self-assembly process of TPR-containing hydrogels allows for encapsulation of molecules. As TPR units can be manipulated to achieve different stabilities and to bind different ligands, it is possible to make them erode under different conditions to trigger cargo release [185]. Enzyme-instructed self-assembly (EISA) represents another approach to construct protein biomaterials that can be applied in targeted delivery, specifically targeting cancer cells [148]. The use of peptide amphiphiles (PA) in biomedicine has recently expanded as drug carriers due to their advantages of unique structures of assemblies, abundant molecular structures, and biological functions [186]. PAs can be formulated to trigger cargo release upon different stimuli. For example, infection sites and tumors present lower pH, which provides a good physical target for controlled release. Hence, pH-sensitive PAs will change their self-assembled structure upon pH changes to ensure controlled release of drugs [187,188]. In addition to changes in temperature [189,190], redox represents another important stimulus that can be applied to trigger specific release of cargo from PAs, thanks to different redox conditions in the intracellular and extracellular compartments, or between healthy and diseased cells [191,192]. The injection of nano-composite hydrogels made of gelatin and laponite containing growth factors and cytokines secreted by stem cells to peri-infarct myocardium has allowed the specific localization and treatment of the infarct area. These hydrogels have been proven to be effective in repairing damaged myocardial tissue by reducing scar area and improving cardiac function [193].

Nowadays, the majority of protein hydrogels used in drug delivery offer poor control over cargo release, which happens either via diffusion, carrier erosion or in response to some of the stimuli we have quoted. Burst release is a common problem in protein hydrogel delivery systems, requiring further efforts to optimize the release profile [194]. Most of the existing systems address specific injection and application areas; however, more work needs to be carried out in the development of novel protein hydrogels that can deliver cargo to specific tissues thanks to molecular recognition. Despite the recognized limitations and challenges with existing protein hydrogels, their prospect in drug and cell delivery is very exciting, as demonstrated by the examples included in this review [195].

### 7.2. Tissue Engineering

The properties of protein hydrogels make them excellent scaffolds for cell culture and tissue engineering. Moreover, the use of protein components allows for the incorporation of sequences associated with both cellular adhesion and growth [196,197]. Depending on the type of tissue and application, hydrogels used as scaffolds can simply physically support tissue formation, relying on the deposition of ECM by cells included in the scaffold, or they can purposely be chosen to trigger specific cell functions or behavior. Natural proteins like collagen and elastin are often involved in fabrication processes ranging from electrospinning to bioprinting to aid in the formation of blood vessels, heart valves, myocardial patches, cartilage, tendons, skin and liver [198,199], although synthetic protein hybrid hydrogels are also employed [200]. Bone and cartilage were some of the first tissues to be substituted or enhanced with the help of metal and composite biomaterials [201,202,203,204]. Slowly but surely, protein hydrogels paved their way into bone tissue engineering, since they allow for site specific injection of cells and growth factors [205]. A recent study has demonstrated an injectable design in which the authors could study the interaction between stroma and hematopoietic stem and progenitor cells, thanks to the use of collagen-coated carboxymethylcellulose microscaffolds. The presence of the collagen layer allowed the establishment and crosstalk between the different cell types [206]. Skin tissue engineering can also benefit from protein hydrogels. Combination of elastin and collagen shows great supporting capacity to create artificial fibroblast-containing dermal patches used in wound healing [46,207]. 

Over the last decade, micro tissues built in microfluidics platforms (organs-on-a-chip) that allow physiological study as well as drug screening, have emerged as a tool for personalized medicine. Some of the most striking tissue developments include microfabricated blood vessels to model vascular transport [208], and pancreatic ductal adenocarcinoma on-a-chip to study endothelial defects [209]. In these two systems, chips take advantages of collagen hydrogels to ensure cell engraftment. Cardiac tissue engineering is one of the most studied organ-on-a-chip platforms for patient-specific studies of physiology and disease, in which fibrinogen and thrombin trap cells in fibrin hydrogels [210,211]. All these cases exemplify the use of protein hydrogels as an aid for tissue engineering, but most of them play a merely passive supportive role. Nonetheless, advanced protein hydrogels can be carefully designed to reproduce the filamentous nature and properties of ECMs and to steer cell behavior as scaffolds for tissue engineering [212]. These emerging applications must be rooted in our understanding of how protein hydrogels interact with cells to uncover how the latter sense and react to the surrounding ECM-a main focus of cell mechanobiology.

With enormous potential for therapeutic applications, several hydrogel formulations have crossed the barriers of in vitro studies and found their way into the market. Some of them are still in the clinical study phases. Among them, some well-defined protein hydrogels have found their way into clinical products applied in tissue engineering and biomedicine. For example, two self-assembling peptides -EAK16 and RADA16- are standardly used in cell regeneration, in surgeries to prevent bleeding from small blood vessels and oozing from capillaries of the parenchyma of solid organs and as scaffolds to promote wound healing after surgeries [213]. Collagen-based hydrogel OP-1^®^, which carries osteogenic protein-1, is currently used in the clinic to treat spinal fusion [160]. There are also other protein hydrogel formulations that currently involved in clinical trials: the combination of gelatin with renal autologous cells is being investigated for the treatment of chronic kidney disease, whereas gelatin-containing fibroblast growth factor is being employed in the treatment of ischemic cardiomyopathy. Native myocardial extracellular matrix is being assessed in the treatment of myocardial infarction [160]. 

### 7.3. Cell Mechanobiology

Over the last 20–30 years, the field of cell mechanobiology has shown that the interplay between mechanical forces and cell biology influences cell behavior, morphogenesis and disease [214,215,216,217]. For instance, proper stem cell differentiation and animal development necessitate mechanical input [91,218,219]; and the mechanics of the nucleus determines the transcriptional state of chromatin [220]. Because of the tight connection between the cytoskeleton and the ECM through cell-surface receptors (e.g., integrins), cells continuously sense the mechanical properties of the ECM and respond exerting traction forces via cell adhesions. These cellular structures are tailored to mechanosense and mechanotransduce mechanical inputs into gene-expression programs that modify cell behavior [91,221,222]. Consequently, the mechanical properties of the ECM determine important cell functions, including adhesion [222,223], proliferation [224], spreading [222], migration [225] and differentiation [91,226]. 

One of the ECM mechanical signatures most commonly studied is stiffness, since its control using synthetic polymers like PDMS or hydrogels like Polyacrylamide (PAAm) is seemingly straightforward. Changes in ECM stiffness modulate cell response both in physiological and pathological scenarios [168,227,228]. Two pioneer examples include the study of the dependence of myocyte striation and subsequent functionality on collagen substrate stiffness [229], as well as integrin clustering and downstream signalling activation induced by collagen substrate stiffening in malignant breast tissue [227]. Additionally, the group of Dennis Discher provided pioneering evidence that ECM stiffness determines cell differentiation [91]. In their experiments Mesenchymal Stem Cells (MSCs) were cultured on PAAm substrates with different elastic moduli. The mechanical stimuli coming from progressively stiffer substrates steered MSCs towards morphologies and RNA transcript profiles similar to neurons, myoblasts and osteoblasts respectively; and it is now well-established that each tissue has a characteristic stiffness and that substrates with a specific elastic modulus are required to generate different cell types [230] (Figure 5a). ECM stiffness influences the maturation of human induced pluripotent stem cell-derived cardiomyocytes, as demonstrated using fibronectin and Matrigel hydrogels attached to surfaces of different stiffness [231]. 

It is important to recognize that most hydrogel systems typically used to study how cells sense and react to ECM stiffness face two limitations. First, most of them behave as linear elastic solids and do not recapitulate the non-linear viscoelastic properties of the ECM and tissues, which typically dissipate energy when strained [232,233] (Figure 5b). Second, the modulation of the mechanical properties in current ECM mimetics is commonly achieved through changes in concentration of building blocks and crosslinking density. As a consequence, these strategies also alter non-mechanical parameters, such as pore size, molecular diffusion, ligand exposure and, oftentimes chemical environment [234,235,236,237,238], which are factors that are also sensed by cells [228,239,240]. This situation potentially leads to intricate convolution and difficult interpretation of experimental results. To address these limitations, the groups of David Mooney and Ovijit Chaudhuri, among others, are paving the way for viscoelastic substrates to understand how cells respond to dissipative cues of the ECM [241,242,243,244,245]. Their work using alginate hydrogels has thoroughly described the impact of stress-relaxation on cell behavior. Viscoelastic properties not only influence cell spreading [243], but are equally important in determining stem cell fate [242,246,247]. Their groundbreaking experiments opened the way for further systems based on different polymers, such as PAAm [248] and PEG [249,250,251]. Although informative, these studies proved challenging to interpret, as the influence of viscoelasticity on cell spreading is not the same in all hydrogel systems. The confounding results obtained may be influenced by the presence of irreversible viscoplasticity in those hydrogels where crosslinking is not covalent, like in the case of alginate and PEG [241]. In this regard, a potential advantage of protein hydrogels over current systems is that dissipation is provided by protein unfolding. The thermodynamic tendency of proteins to go back to their native status will difficult the formation of new bonds that lead to plasticity, hence making protein hydrogels viscoelastic. As discussed above, the intrinsic mechanical stability of proteins can be chosen to produce hydrogels with tailored mechanical properties [120,144]. Thus, it is possible that protein hydrogels can be used to modulate viscoelasticity while preserving non-mechanical properties, and their use in cell mechanobiology experiments may provide insights into how cell sense energy dissipation of the ECM.

**Figure 5 nanomaterials-11-01656-f005:**
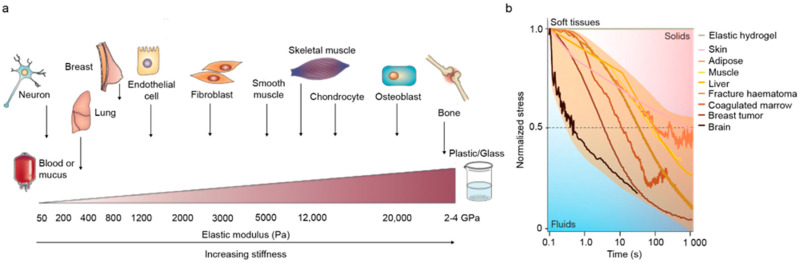
Biological tissues are viscoelastic. (**a**) All tissues are exposed to mechanical forces. Each cell type is specifically tuned to fit the stiffness of the specific tissue in which it resides. (**b**) Stress relaxation tests on the indicated tissues. Under constant strain purely elastic hydrogels do not relax over time (grey line), whereas almost all tissues in the body do. Adapted with permission from [215] Copyright, Springer Nature, 2019 and [224]. Copyright, Springer Nature, 2020.

Despite the aforementioned challenges, mechanistic studies using linear elastic ECM mimetics have uncovered several mechanotransduction signalling pathways in cells, such as Rho signalling and small GTPAses [252], TGF-β [253] and the Hippo pathway [254,255]. YAP (Yes-associated protein) and TAZ (transcriptional coactivator with PDZ-binding motif) are members of the Hippo pathway that play a pivotal role in regulation of cell proliferation and growth and are involved in tumor suppression [254,256]. Dephosphorylated YAP/TAZ translocate to the nucleus to regulate gene expression programs modulating multiple essential biological processes [254]. Monitoring the activity of YAP/TAZ has been proposed as proxy of how cells interact with biomaterials [257]. When cells grow on stiff elastic substrates, YAP/TAZ are activated and translocated to the nucleus to modulate gene expression (e.g., ECM synthesis, deposition and remodelling), whereas on soft elastic substrates YAP/TAZ mostly remain in the cytosol [258,259]. How YAP/TAZ react to ECM energy dissipation is still being defined [242,248,249,250,260,261,262]. In the future, it will be interesting to study the activity of YAP/TAZ when cells are cultured on protein hydrogels of controlled dissipative properties.

## 8. Limitations of Protein Hydrogels

Naturally-derived protein hydrogels originally faced two main setbacks concerning the obtaining of the protein building blocks and the mechanical properties of the resulting hydrogels [195,263]. Traditionally, the extraction of proteins from tissues required lengthy and technically complicated processes, often involving the use of strong acids [264]. Thanks to advances in recombinant DNA technology and in molecular biology it is now possible to produce larger amounts of protein using expression systems like bacteria and yeasts [265,266], which bypasses the original limitation. We need to stress that the design of synthetic proteins requires expertise in protein engineering, as well as the ability to perform not so straightforward purification procedures that nevertheless are becoming more and more common thanks to the increasing availability of convenient commercial solutions. On the other hand, the mechanically labile properties of natural protein hydrogels can be enhanced by combining these proteins with synthetic polymers, which expands their range of applications. For example, the elastic moduli of collagen hydrogels can be increased by stiffening interconnected collagen fibers with varied amounts of poly(ethylene glycol) di(succinic acid N-hydroxysuccinimidyl ester) [267,268]. Furthermore, production of recombinant polyproteins allows researchers to crosslink hydrogels with improved mechanical responses depending on the building block of choice [122], although there is still room for improvement in the array of mechanical behaviors of protein hydrogels.

A current limitation of protein hydrogels is the purification yield. Typical protocols only produce a few mg of protein, which hinders the use of protein hydrogels in high throughput screenings and makes scaling up protein production difficult. Although the need remains to increase the production yield of protein building blocks, we acknowledge that both prokaryotic and eukaryotic expression systems have undergone remarkable improvements over the years, making them more user-friendly and adaptable to the needs of specific proteins [266,269]. Furthermore, the adaptation of tensile testers to very low volume hydrogels bypasses the limitation of protein purification yield allowing mechanical characterization of samples of only a few microliters [154,270].

## 9. Conclusions

Protein-based hydrogels constitute versatile tools that can overcome current limitations of typical linear elastic ECM mimetics employed in mechanobiology and tissue engineering. The mechanical properties of protein hydrogels are rooted in the nanomechanics of their building blocks, which can be modified with subtle changes while in principle preserving protein concentration and crosslinking density [110,122,271]. In fact, proof-of-principle observations showing that it is possible to modulate the mechanical properties of protein hydrogels at the same protein concentration and crosslinking density, resulting in changes in cell behavior, have been obtained using hydrogels that are responsive to chemical environment via redox status and pH [119,128,144]. Together with the ability to incorporate protein functions into hydrogels, protein hydrogels represent a gateway to engineer ECM-mimicking substrates whose mechanical properties can be specifically controlled for applications ranging from tissue engineering to interrogation of cell mechanosensing.

## Figures and Tables

**Figure 1 nanomaterials-11-01656-f001:**
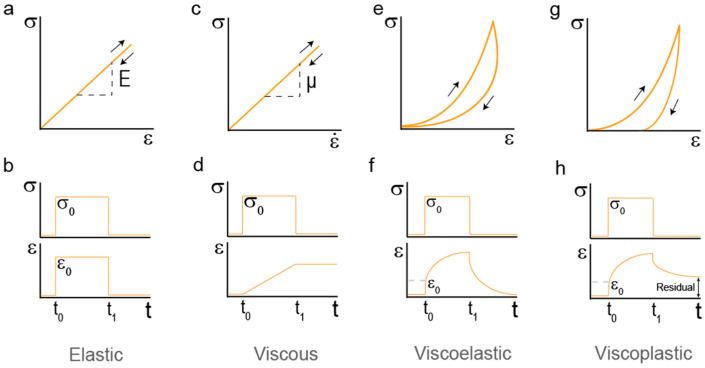
Characterization of hydrogel mechanics (**a**) Stress (σ)-strain (ε) proportional relationship of a classical linear elastic solid. The slope E is the elastic modulus of the material. (**b**) Stress and strain histories for a linear elastic solid. A load σ_0_ is applied at t = t_0_ and removed at t_1_. This load generates a simultaneous strain. (**c**) Stress–strain rate plot for a linear viscous fluid shows a proportional relationship between them. The slope μ is the viscosity of the material. (**d**) Mechanical response of a linear viscous fluid. Representation of stress histories. Stress σ_0_ is applied at t = t_0_ and removed at t_1_. This load generates a progressive strain response. (**e**) Stress–Strain measurement of a viscoelastic solid. Energy dissipation appears in the form of hysteresis. (**f**) Creep response of a viscoelastic solid. Strain increases while stress is kept constant. When stress is removed, strain drops to zero in a time-dependent fashion. (**g**) Stress–strain measurement of a viscoplastic solid. Energy dissipation appears in the form of hysteresis. Plasticity is represented as the strain never going back to zero after load is completely removed. (**h**) Creep response of a viscoplastic solid. Strain increases while stress is kept constant. The presence of plastic behavior is highlighted by residual strain after recovery from stress.

**Figure 2 nanomaterials-11-01656-f002:**
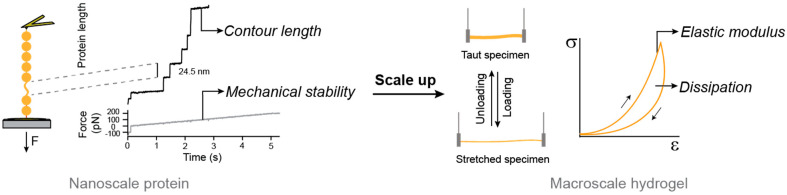
Bottom-up approach to engineer the mechanical properties of protein hydrogels. Protein nanomechanics can be studied through single-molecule techniques such as atomic force microscopy represented on the left of the figure. Force ramp experiments trigger the mechanical unfolding of individual protein domains in a polyprotein building block. Changes in contour length and the force at which the unfolding takes places determine the mechanical behavior of the protein domains. Polyproteins of interest can be used to produce protein hydrogels. Their mechanical response in terms of elastic modulus and energy dissipation can be measured using tensile tests including loading-unloading cycles. The mechanical behavior of the hydrogel depends on that of its building blocks.

**Figure 3 nanomaterials-11-01656-f003:**
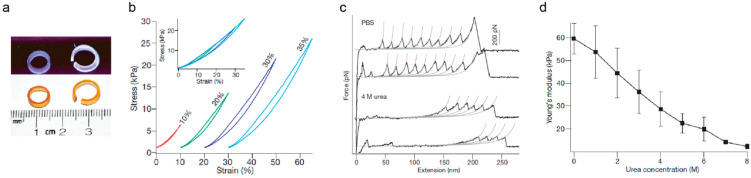
Mechanical properties of protein-based biomaterials. (**a**) Photographs of molded rings built from protein-based biomaterials under white light (bottom panel) and ultraviolet illumination showing fluorescence derived from dityrosine crosslinks (top panel). (**b**) Representative stress–strain curves of protein-based biomaterials (**c**) Force–extension AFM curves of single protein building blocks in PBS and in urea, which triggers protein unfolding. The long featureless spacers observed in force-extension curves in urea largely correspond to the stretching of mechanically labile, unfolded domains. The unfolding force of domains that remain folded in urea is also significantly reduced. Grey lines are fits to the worm-like chain model of polymer elasticity. (**d**) Young’s modulus of protein-based biomaterial is modulated by urea. The conversion of folded domains into unfolded sequence leads to the dramatic decrease in Young’s modulus of the biomaterials in a urea-concentration-dependent manner. Error bars indicate standard deviation of the data. Adapted with permission from [81]. Copyright, Springer Nature, 2010.

## Data Availability

The data presented in this study are available on request from the corresponding author.

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
