# Peer review of "Protein Hydrogels: The Swiss Army Knife for Enhanced Mechanical and Bioactive Properties of Biomaterials"

_nanomaterials, 2021, doi:10.3390/nano11071656_

Round 1

Reviewer 1 Report

The paper presents a comprehensive review of the current state of the art in the field of protein hydrogels including homopolymeric and heteropolymeric hydrogels. Two major properties of this type of hydrogels are considered, namely mechanical characteristics and biological activity. The review is a valuable source of information on the selected topic and provides adequate references to related and previous work.

I have the following comments:

  1. Section "Mechanical properties of hydrogels" contains too much basic information on the mechanical properties of solids. This makes this section look like a student's textbook, not a scientific review. Sections 2.1, 2.2, 2.3 should be rewritten or even excluded from the review.
  2. Instead of basic information on the mechanical properties of solids, much more information is needed on the composition and structure of natural and engineered protein-based hydrogels and its influence on the mechanical properties of hydrogels. SAXS(USAXS) and SANS (USANS) data on the structure of these protein gels should be discussed.
  3. Surprisingly, I have not found any data concerning cosmeceutic applications of protein hydrogels. Only advanced applications (cell and gene delivery, tissue engineering etc.) are considered, while traditional ones are largely ignored.

I'd like to recommend major revision of the paper aiming at addressing the issues mentioned above.

Author Response

Response to Reviewer 1 Comments

Title: Protein hydrogels: the Swiss army knife for enhanced mechanical and bioactive properties of biomaterials

Manuscript ID: nanomaterials-1252543

The paper presents a comprehensive review of the current state of the art in the field of protein hydrogels including homopolymeric and heteropolymeric hydrogels. Two major properties of this type of hydrogels are considered, namely mechanical characteristics and biological activity. The review is a valuable source of information on the selected topic and provides adequate references to related and previous work.

We would like to thank the reviewer for their appreciation of the manuscript as well as for the careful evaluation of our work and for the suggested clarifications. Following their suggestions, we now have edited the manuscript to clarify some parts and to include the required new information. As a result of this revision, we believe that our manuscript has been greatly improved. Please, see below a point-by-point response to the reviewer’s comments.

Point 1: Section "Mechanical properties of hydrogels" contains too much basic information on the mechanical properties of solids. This makes this section look like a student's textbook, not a scientific review. Sections 2.1, 2.2, 2.3 should be rewritten or even excluded from the review.

Response 1: We agree with the reviewer and we have tried to modify this section accordingly. We have removed information that is not pertinent to a scientific review. We have synthesized the information in sections 2.1, 2.2, 2.3 and 2.4 into just revised section 2, where the essential features and mechanical behavior of elastic, viscoelastic and viscoplastic materials are described. These are important concepts to understand the use of hydrogels in cell mechanobiology, which can be helpful to many cell biologists. We have also modified figure 1 to adapt it to the new version of the text.

Point 2: Instead of basic information on the mechanical properties of solids, much more information is needed on the composition and structure of natural and engineered protein-based hydrogels and its influence on the mechanical properties of hydrogels. SAXS(USAXS) and SANS (USANS) data on the structure of these protein gels should be discussed.

Response 2: We thank the reviewer for this suggestion. We have incorporated a new section 3, named “Composition and structural properties of hydrogels”, where SAXS, SANS and how the structural properties of hydrogel building blocks influence hydrogel mechanical behavior are discussed.

Point 3: Surprisingly, I have not found any data concerning cosmeceutic applications of protein hydrogels. Only advanced applications (cell and gene delivery, tissue engineering etc.) are considered, while traditional ones are largely ignored.

Response 3: We have now included examples of protein hydrogels employed in cosmeceutical applications. These examples can be found in the first three paragraphs of new section 7.

Reviewer 2 Report

Overall, the authors gave good presentation of hydrogels with controlled mechanical properties and their biomedical applications. However, I suggest the authors could consider the following issues:

Swiss army knife definitely is cool, but it is really difficult for me and the potential readers to immediate grip the ideas why Swiss army knife come out in the title? Throughout the manuscript, Swiss army knife does not appear second time. 

In the section of "drug and cell delivery", I do not think the authors give appropriate summary of the current progress in using hydrogel in delivery of drugs and cells, and also highlight the advantages of using hydrogel as delivery vehicles. The examples that the authors listed are basically controlled release of drugs in the administered site rather than "delivering to drug to the desirable site".

Last, the authors at least should give several examples of hydrogels that already approved  in clinical applications so as to emphasize the hydrogel in biomedical applications is convincing. As well, what is the current bottlenecks? how scientist strived to overcome these bottlenecks?

Author Response

Response to Reviewer 2 Comments

Title: Protein hydrogels: the Swiss army knife for enhanced mechanical and bioactive properties of biomaterials

Manuscript ID: nanomaterials-1252543

Overall, the authors gave good presentation of hydrogels with controlled mechanical properties and their biomedical applications. However, I suggest the authors could consider the following issues.

We would like to thank the reviewer for their insights on the different applications of hydrogels and for all their comments aimed at making the manuscript better. We have incorporated them to the original version and we believe the review has greatly improved.

Point 1: Swiss army knife definitely is cool, but it is really difficult for me and the potential readers to immediate grip the ideas why Swiss army knife come out in the title? Throughout the manuscript, Swiss army knife does not appear second time. 

Response 1: We agree on this title being a little difficult to grasp. Since we thought of it as a catchy way to indicate the versatility of protein hydrogel systems, we have now clarified the meaning of this analogy in the introductory paragraph of the manuscript.

Point 2: In the section of "drug and cell delivery", I do not think the authors give appropriate summary of the current progress in using hydrogel in delivery of drugs and cells, and also highlight the advantages of using hydrogel as delivery vehicles. The examples that the authors listed are basically controlled release of drugs in the administered site rather than "delivering to drug to the desirable site".

Response 2: We thank the reviewer for this suggestion. We have tried to provide more examples on protein hydrogel-mediated drug delivery in new section 7.1, including targeted strategies in response to changes in external stimuli, such as pH and temperature. We have also discussed the promise of targeted delivery based on molecular recognition. Unfortunately, we have not found protein hydrogel systems that achieve this goal, so we would appreciate to learn about any specific paper regarding this topic that the reviewer may be aware of.

Point 3: Last, the authors at least should give several examples of hydrogels that already approved in clinical applications so as to emphasize the hydrogel in biomedical applications is convincing. As well, what is the current bottlenecks? how scientist strived to overcome these bottlenecks?

Response 3: We have taken into consideration the suggestions of the reviewer and have included several examples of protein hydrogels employed in the clinic, both focused on cosmeceutical and tissue engineering applications. Examples can be found in red in lines 517-520 and 650-663 of the marked version of the manuscript. Regarding the presence and overcoming of bottlenecks, we have expanded the section “Limitation of protein hydrogels” and described several of the drawbacks protein hydrogels have experienced as well as some strategies to overcome them.

Round 2

Reviewer 1 Report

The authors have addressed most of my comments. The paper is now suitable for publication.